# Early Supplementation with Branched-Chain Amino Acids Ameliorates Lipid Retention in Aortic Valves of ApoE-Knockout Mice

**DOI:** 10.3390/ijms262311259

**Published:** 2025-11-21

**Authors:** Daria Shishkova, Anastasia Kanonykina, Egor Kondratiev, Arina Tyurina, Alexandra Morozova, Alena Poddubnyak, Anna Sinitskaya, Maxim Sinitsky, Victoria Markova, Anastasia Lazebnaya, Leo Bogdanov, Alexander Stepanov, Susanna Agalaryan, Anton Kutikhin

**Affiliations:** Department of Experimental Medicine, Research Institute for Complex Issues of Cardiovascular Diseases, 6 Barbarash Boulevard, 650002 Kemerovo, Russia; shidk@kemcardio.ru (D.S.); kanoau@kemcardio.ru (A.K.); kondea@kemcardio.ru (E.K.); turiae@kemcardio.ru (A.T.); moroag@kemcardio.ru (A.M.); poddao@kemcardio.ru (A.P.); cepoav@kemcardio.ru (A.S.); sinimu@kemcardio.ru (M.S.); markve@kemcardio.ru (V.M.); lazeai@kemcardio.ru (A.L.); bogdla@kemcardio.ru (L.B.); stepad@kemcardio.ru (A.S.); agalsa@kemcardio.ru (S.A.)

**Keywords:** proteinogenic amino acids, human serum albumin, magnesium citrate, atherogenesis, calcification, anti-atherogenic effect, anti-calcific effect, aortic valve, ionized calcium, serum calcification propensity

## Abstract

Previous studies suggested a certain efficiency of proteinogenic branched-chain amino acid (BCAA) and magnesium supplementations in reducing cardiovascular risk and increasing quality of life. This investigation assessed the anti-atherogenic and anti-calcific effects of BCAA (55 mg/day, corresponding to a human equivalent dose of 13.5 g/day) and magnesium citrate (MgCit, 1.85 mg/day, corresponding to a human equivalent dose of 450 mg/day) intake in male and female ApoE-knockout mice, with the treatment initiation at either 1, 3, or 6 months of age. At the 12-month time point, lipid retention and calcium deposition in the aortic valve, lipid burden in the aorta, and serum ionized calcium were evaluated. The early BCAA intake (from 1/3 to 12 months of age) significantly reduced lipid retention in the aortic valve, whilst MgCit decreased ionized calcium. Both of these protective effects were higher in male than in female mice. Furthermore, it was tested whether human serum albumin (HSA) or MgCit can be applied to decrease the serum calcification propensity in 100 patients with myocardial infarction. A dual supplementation with HSA and MgCit reduced serum calcification propensity in 68% of cases. Collectively, these results highlight the potential benefits of BCAA/HSA and magnesium supplementations for cardiovascular prevention and justify further clinical trials in this regard.

## 1. Introduction

Significant advances in the pharmacological prevention of major adverse cardiovascular events have been made in recent decades [1,2,3]. A variety of lipid-lowering medications such as statins, fibrates, cholesterol absorption inhibitors, omega-3-acid ethyl esters, adenosine triphosphate citrate lyase inhibitors, nicotinic acid, and proprotein convertase subtilisin/kexin type 9 inhibitors showed an efficiency in correcting the lipid profile of patients with atherosclerosis [4,5], and novel targeted hypolipidemic therapies, including small interfering RNA molecules, angiopoietin-like 3, 3/8, and 4 inhibitors, apolipoprotein C-III inhibitors, fibroblast growth factor-21 analogs, and microsomal triglyceride transfer protein inhibitors, are currently undergoing clinical trials [6,7,8]. Yet cardiovascular diseases still remain the most common cause of death worldwide, with 626 million prevalent cases and 19.2 million deaths documented in 2023, although around 80% of cardiovascular disease burden is attributable to modifiable risk factors and therefore can be reduced proactively by implementing low-cost and broadly available preventive approaches [9]. Dietary supplements, such as antioxidant vitamins, mono- and polyunsaturated fatty acids, short-chain fatty acids, polyphenols, phytosterols, curcuminoids, carotenoids, allicin, berberine, coenzyme Q10, red yeast rice, and fiber, have been suggested as promising agents for complementing and enhancing the lipid-lowering effects of the abovementioned drugs [10,11,12,13,14,15,16].

Among other nutraceuticals, a proteinogenic branched-chain amino acid (BCAA: leucine, isoleucine, and valine) supplement was shown to attenuate atherosclerotic burden, normalize lipid profiles, hinder immune cell infiltration, reduce levels of pro-inflammatory cytokines (tumor necrosis factor alpha, interleukin-1β, interleukin-6, and monocyte chemoattractant protein-1), and normalize the gut microbiota in hyperlipidemic mice [17]. A treatment with BCAA also diminished the production of interleukin-6, interleukin-8, and cyclooxygenase-2 and preserved telomere length in lipopolysaccharide-activated macrophages [18] and cancer cells [19]. The administration of BCAA augmented albumin synthesis in primary hepatocyte cultures [20] and in rats with liver injury [21], and recent meta-analyses [22,23,24] confirmed the efficiency of BCAA supplementation for increasing serum albumin in patients with liver cirrhosis [22], hepatocellular carcinoma [23], and other diseases requiring hepatic intervention [24]. The cardioprotective effects of albumin include the binding of excessive fatty acids [25,26,27], ionized calcium [28], and free radicals [29,30,31], the direct suppression of cytokine production [32,33], the inhibition of platelet aggregation [34], and multiple anticoagulant effects such as a reduced clot firmness, prolonged clot formation, and enhanced antithrombin activity [34]. Low serum albumin has been associated with venous thromboembolism [35,36], myocardial infarction [35,36,37,38], stroke [35,37], and cardiovascular death [39].

Magnesium supplementations, such as magnesium citrate (MgCit), demonstrated the potential to inhibit vascular calcification in rats with adenine-induced chronic renal failure [40], probably through the reduction in parathyroid hormone release [41] and by the induction of anti-inflammatory effects as measured by decreased C-reactive protein [42] and tumor necrosis factor alpha [43]. Moreover, magnesium intake improved flow-mediated dilation [44,45], reduced the blood pressure [46,47,48,49], decreased low-density lipoprotein cholesterol and triglycerides [50,51], and increased high-density lipoprotein cholesterol [51,52,53], altogether lowering cardiovascular risk [54]. Likewise, hypomagnesemia is associated with cardiovascular mortality in patients with end-stage renal disease [55,56,57,58]. The advantageous effects of magnesium can be explained by its antagonistic action on ionized calcium, whose excess is associated with major adverse cardiovascular events [59] and cardiovascular death [60]. Collectively, these data suggest the potential utility of BCAA and magnesium supplementation for the prevention of atherosclerosis and reduction of cardiovascular risk.

Previous studies indicated sex differences in the metabolism of BCAA [61,62] and responses to BCAA and magnesium supplements [63,64], presumably because the absorption and distribution of magnesium [65] and catabolism of BCAA [66] are affected by estrogen levels. Serum levels of BCAA had the strongest associations with body mass index in males and with waist circumference and waist-to-height ratio in females [62]. Along similar lines, BCAA supplementation better improved fatigue perception in females than in males, whilst the latter primarily benefitted from higher strength gains [63]. Magnesium intake showed higher effects on bone mineral density in females than in males [64]. These findings suggest the need for the evaluation of sex-specific effects in the studies assessing the favorable effects of BCAA and magnesium supplementation.

Having considered the potential anti-atherosclerotic effects of BCAA and anti-calcific effects of magnesium supplementations, this study had an objective to evaluate whether these treatments reduce lipid and calcium deposition in the aortic valve (AV) and lipid burden in the aorta of hyperlipidemic (ApoE-knockout) mice. To further explore the clinical benefits of human serum albumin (HSA) and MgCit, the study has also tested their ability to reduce serum calcification propensity in patients with ST-segment elevation myocardial infarction (STEMI).

## 2. Results

For the extensive investigation of the anti-atherogenic and anti-calcific effects of BCAA and MgCit, this study included three independent experiments with an initiation of BCAA (55 mg/day, which corresponds to a human equivalent dose of 13.5 g/day) or MgCit (1.85 mg/day, which corresponds to a human equivalent dose of 450 mg/day) supplementation in ApoE-knockout mice at 1, 3, and 6 months of age. The human equivalent doses were calculated by dividing the dose for a mouse by 12.3 (a coefficient established by the Food and Drug Administration). At 12 months of age, all animals were euthanized, with the subsequent evaluation of lipid retention in the AV and aorta through Oil Red O staining and the assessment of calcium deposition in the AV through Alizarin Red S staining. The initiation of BCAA treatment at 1 and 3 months reduced the lipid burden in the AV by ≈30% in comparison with mock-treated mice (5.38% (4.39–6.78%) vs. 7.54% (6.72–9.22%), respectively, at 1 month; 7.34% (6.03–12.19%) vs. 10.49% (8.89–16.27%), respectively, at 3 months; Figure 1 and Figure 2, Appendix A). Yet this effect was lost if the BCAA intake started at 6 months of age (7.38% (4.60–9.76%) and 6.47% (5.12–8.12%) in BCAA- and mock-treated groups, respectively; Figure 1 and Figure 2, Appendix A). Such effects of BCAA treatment were more pronounced in male mice (5.33% (4.42–6.78%) vs. 7.54% (7.12–8.75%) in BCAA- and mock-treated groups, respectively, at 1 month; 11.24% (7.57–12.62%) vs. 14.82% (10.89–20.22%), respectively, at 3 months; 4.66% (3.14–6.80%) vs. 7.02% (6.25–8.24%), respectively, at 6 months; Figure 1 and Figure 2, Appendix A) than in female mice (5.81% (4.30–6.93%) vs. 7.54% (5.98–10.14%), respectively, at 1 month; 6.62% (5.75–7.11%) vs. 9.26% (6.95–10.18%), respectively, at 3 months; 9.74% (8.18–12.40%) vs. 5.65% (4.84–8.07%), respectively, at 6 months; Figure 1 and Figure 2, Appendix A). In total, BCAA intake decreased the lipid burden in the AV by ≈20% compared with mock-treated mice (6.73% (4.81–8.47%) vs. 8.46% (6.46–11.05%), respectively), with a ≈25% reduction in male mice (6.50% (4.42–8.38%) vs. 8.84% (7.24–14.82%) in BCAA- and mock-treated groups, respectively) and only a ≈15% reduction in female mice (6.85% (5.74–8.95%) vs. 7.91% (5.88–9.89%), respectively; Figure 1 and Figure 2, Appendix A). In contrast to BCAA, the intake of MgCit inconsistently affected lipid deposition in the AV in comparison with the mock treatment (7.94% (6.72–10.16%) vs. 8.46% (6.46–11.05%) in the pooled sample, respectively; Figure 1 and Figure 2, Appendix A). In the pooled sample of male mice, MgCit showed an insignificant protective effect (7.84% (6.63–10.00%) vs. 8.84% (7.24–14.82%) in MgCit- and mock-treated groups, respectively) but the association was inverse in the female mice (8.70% (6.98–11.39%) vs. 7.91% (5.88–9.89%), respectively; Figure 1 and Figure 2, Appendix A).

Neither the BCAA nor the MgCit intake was able to consistently reduce the lipid burden in the aorta regardless of the treatment duration (Figure 3 and Figure 4), although BCAA supplementation reduced it by ≈50% in male mice if the treatment was started at 1 month of age (12.27% (7.43–18.91%) vs. 23.36% (18.68–26.74%) in BCAA- and mock-treated groups, respectively; Figure 3 and Figure 4). However, this effect was not retained in the pooled sample (16.81% (10.78–25.25%) vs. 17.66% (12.18–25.45%) in both sexes; 23.78% (13.14–30.77%) vs. 24.14% (17.94–29.68%) in male mice; Figure 3 and Figure 4) and was not found in female mice (12.77% (9.29–17.55%) vs. 12.32% (8.60–16.51%), respectively, in BCAA- and mock-treated groups in the pooled sample; 10.07% (2.66–12.67%) vs. 8.79% (3.42–11.51%) if the treatment was initiated at 1 month of age; Figure 3 and Figure 4).

Calcium deposition in the AV also was not affected by BCAA or MgCit supplementation (4.20% (2.19–7.60%) vs. 3.82% (1.97–7.35%) vs. 3.86% (2.42–6.28%), respectively, in BCAA-, MgCit-, and mock-treated groups in the pooled sample; Figure 5 and Figure 6, Appendix A). Again, BCAA- or MgCit-fed male mice had an insignificant reduction in the amount of calcium (3.58% (1.43–6.33%) vs. 2.98% (1.37–4.25%) vs. 3.85% (2.56–6.37%) in BCAA-, MgCit-, and mock-treated groups, respectively; Figure 5 and Figure 6, Appendix A), whilst an inverse association was found in the pooled sample of female mice (4.46% (3.07–10.39%) vs. 6.22% (2.79–9.28%) vs. 3.86% (2.36–6.38%) in BCAA-, MgCit-, and mock-treated groups, respectively; Figure 5 and Figure 6, Appendix A). Collectively, these results indicate the sex-specific effects of BCAA and MgCit intake, which were more efficient in reducing the valvular and vascular lipid and calcium burdens in male but not female mice.

The treatment with MgCit and BCAA lowered the level of serum ionized calcium (1.29 (1.24–1.34) vs. 1.32 (1.25–1.37) vs. 1.34 (1.29–1.40) mmol/L in the pooled sample of MgCit-, BCAA-, and mock-treated mice, respectively; Figure 7). In concert with a reduced AV calcification, the indicated reduction in serum ionized calcium was more pronounced in the pooled sample of male mice (1.29 (1.25–1.34) vs. 1.30 (1.24–1.38) vs. 1.35 (1.29–1.39) mmol/L in MgCit-, BCAA-, and mock-treated mice) than in female mice, where it did not reach statistical significance (1.29 (1.24–1.37) vs. 1.32 (1.27–1.37) vs. 1.32 (1.26–1.40) mmol/L, respectively; Figure 7). Although such a decrease in the serum ionized calcium in the total sample and in male mice was statistically significant exclusively in the MgCit group, a similar trend has been observed in BCAA-treated mice (Figure 7).

To further investigate the impact of BCAA and MgCit on calcium metabolism, ex vivo serum calcification propensity was evaluated in 100 patients with STEMI by calculating the difference in the optical density at a 620 nm wavelength (normalized OD_620_) between the paired serum samples with and without Ca/P supersaturation. The addition of HSA (+10 mg/mL, equivalent to the interquartile serum albumin in the population), MgCit (+90 µg/mL, equivalent to 450 mg/day), or both indicated solutions resulted in a statistically significant reduction in OD_620_ (Figure 8). MgCit and HSA supplementation decreased the normalized OD_620_ in 57/100 (57%) and 55/100 (55%) patients, respectively (Figure 8). The combined ex vivo treatment with MgCit and HSA led to a decline in serum calcification propensity in 68/100 (68%) Ca/P-supersaturated but only in 7/100 (7%) control serum samples (*p* < 0.0001), and anti-calcific effects of MgCit and HSA were found exclusively in the Ca/P-supersaturated but not the control serum (Figure 8), confirming specific actions of both treatments.

The study pipeline and the main findings are illustrated in the summary figure (Figure 9).

## 3. Discussion

Randomized clinical trials and meta-analyses showed that BCAA supplementation has a number of beneficial effects, including the stimulation of muscle protein synthesis [67,68,69], reduction in muscle soreness [70,71,72,73], and inhibition of muscle loss [74,75,76], in particular in the elderly [67,74] and patients with liver cirrhosis suffering from hypoalbuminemia [68,69,75,76]. Several studies also reported that BCAA intake improved survival in mice [77] and humans [78], in particular patients with liver cirrhosis [22,79]. Magnesium supplementations, such as MgCit, have also shown efficiency in improving the quality of life of stressed adults [80] and of patients with a major depressive disorder [81] or type 2 diabetes mellitus [82], potentially by exerting anti-inflammatory effects [83]. Low serum albumin, whose synthesis is stimulated by BCAA supplementation [20,21,22,23,24], has been associated with venous thromboembolism [35,36], major adverse cardiovascular events [35,36,37,38], and cardiovascular death [39,84]. Along similar lines, low serum magnesium correlated with arterial hypertension [85], coronary artery disease [86,87,88], major adverse cardiovascular events [89], and cardiovascular death [55,56,57,58]. This study examined whether BCAA (55 mg/day, corresponding to a human equivalent dose of 13.5 g/day) or MgCit (1.85 mg/day, corresponding to a human equivalent dose of 450 mg/day) intake can reduce the lipid and calcium burden in the AVs and blood vessels of ApoE-knockout mice. As the current guidelines for animal studies require equal numbers of male and female mice [89,90,91,92,93,94,95], and having considered the estrogen-dependent effects of BCAA [61,62,63,66] and magnesium [64,65] supplementations, both pooled and sex-specific analyses have been performed. Furthermore, it was assessed whether the timing of the treatment initiation (1, 3, and 6 months of age, which corresponds to 16–19, 20–34, and 35–44 years of age in humans) affects its outcome at 12 months of age (corresponding to 60–74 years of age in humans).

An early start of BCAA supplementation (1 or 3 months of age) is associated with a reduced lipid deposition in the AV, whereas MgCit can be efficient in the control of serum ionized calcium. Both indicated protective effects were higher in male mice in comparison with female mice, in keeping with the previous reports [62]. Our results are in concord with a previous study on ApoE-knockout mice which found that treatment with BCAA reduces the area of atherosclerotic lesions, presumably due to a decrease in serum triglycerides and low-density lipoprotein cholesterol, lowered production of pro-inflammatory cytokines, hindered macrophage migration, and increased bile acid excretion [17]. Likewise, the oral intake of magnesium by ApoE-knockout mice was associated with a decreased plaque area as well as reduced total cholesterol and triglycerides [96]. Although neither BCAA nor MgCit affected lipid retention in the aorta or calcium deposition in the AV, the measurement of lipid content in the AV seems to be a more sensitive technique to quantify the lipid burden in ApoE-knockout mice [97,98,99,100,101,102,103]. Therefore, BCAA supplementation (≈12.5 g/day) can hold promise in inhibiting the development of atherogenesis, particularly in patients with a high cardiovascular risk such as those with familial hypercholesterolemia, which is mirrored by the knockout of the *Apoe* gene and a rise in both total and low-density lipoprotein cholesterol in the respective mouse models. An additional oral supplementation with MgCit or other magnesium preparations (≈400–450 mg/day) might be suggested as a promising option to reduce ionized calcium from a highest (i.e., risk quartile) to the lowest (i.e., protective) quartile [59,60].

Moreover, the supplementation with HSA, which is synthesized from BCAA, and MgCit significantly decreased serum calcification propensity, with a 68% efficiency of the combined treatment as compared with 7% in the group without serum supersaturation with calcium and phosphate. In previous studies, serum calcification propensity has been associated with myocardial infarction [104,105] and cardiovascular death [106] regardless of comorbid cardiovascular risk factors, including glomerular filtration rate [104,105,106], with major adverse cardiovascular events in patients with chronic kidney disease (from stage 2 to stage 4) [107], and with cardiac mortality in kidney transplant recipients [108]. Furthermore, serum calcification propensity was associated with the progression of calcific aortic stenosis [109], coronary artery calcification [110], aortic calcification [111], and a severe phenotype of pseudoxanthoma elasticum accompanied by an accelerated vascular calcification [112]. As our protocol for defining serum calcification propensity fully corresponded to those applied earlier in calciprotein monomer and calciprotein particle research [113,114,115], the results of this study suggest the potential utility of BCAA and MgCit for the inhibition of serum calcification propensity in various patient categories, including those with cardiovascular or chronic kidney disease. In addition to BCAA, HSA preparations can also be used to increase serum albumin and lessen serum calcification propensity, although they demand intravenous injections instead of oral intake, therefore restricting their prolonged application (e.g., in mice). Yet experimental studies [20,21] and meta-analyses [22,23,24] convincingly demonstrated that treatments with BCAA augment the biosynthesis of albumin by hepatocytes, justifying the use of BCAA supplementations for elevating serum albumin levels, in particular in patients with liver cirrhosis.

Among the advantages of our study is the equal representation of male and female mice, which allowed us to find the sex-specific gain in BCAA- and MgCit-related effects. The reduction in lipid retention at an early supplementation with BCAA was higher in the AVs and aortas of male mice as compared to female mice, and a similar trend was observed with regard to the lowering of the serum ionized calcium. A comparison of the efficiency of BCAA and MgCit intake at ascending age (1, 3, and 6 months of age) enabled us to show the benefits of early treatment initiation (from 1 to 3 months of age, corresponding to 6 to 34 years of age in humans) over a late start of the supplementation (from 6 months of age, corresponding to ≥35 years of age in humans). Relatively large samples (135 mice per sex, 90 mice per treatment group, 90 mice per treatment initiation time point) and the relatively high proportion of mice who survived until the 12-month study endpoint and have been included into the analysis (222 out of 270, ≈82%) ensured a sufficient statistical power. A lipid-rich diet has not been applied in order to better conform to the real-world clinical setting of hyperlipidemia without an excessive increase in serum atherogenic lipids (i.e., crude fat content was restricted to 6%). The selected dosages were in accord with the recommended human equivalent doses of BCAA (from 4 to 20 g/day, with an average around 12–14 g/day) and MgCit (≈400–450 mg/day) and were slightly (≈10%) excessive to correct for the amount of undrunk water with the nutraceuticals. Furthermore, BCAA and MgCit showed an efficiency in reducing the serum calcification propensity, a novel cardiovascular risk factor, in patients with STEMI, thus reinforcing the justification of the randomized controlled trials to challenge anti-atherogenic, anti-calcific, and cardioprotective effects of the indicated supplementations. The study results can also be replicated on another strain of hyperlipidemic mice (e.g, Ldlr-knockout), utilizing a lipid-rich diet (≈22% crude fat content), or using an earlier study endpoint (e.g., 9 months of age, which corresponds to 45–59 years of age in humans) that can be more sensitive as compared to the 12-month endpoint applied in our investigation. Another direction is to test the efficiency of BCAA and MgCit supplementation in hyperlipidemic mice with comorbid conditions, such as chronic kidney disease (e.g., as a result of a partial nephrectomy), obesity, or diabetes mellitus, including mouse models with an introduced negative genetic background. The absence of an extended biochemical analysis (e.g., measurement of estrogen or androgen levels) due to the insufficient amounts of serum is the limitation of our study and can also be overcome in the following investigations. The effects of magnesium and BCAA/HSA supplementation on serum iron, zinc, or other electrolytes require further studies both in male and female mice. Yet both of these treatments are generally considered as relatively safe and do not bear major side effects if not exceeding the recommended dosages.

To summarize, BCAA and MgCit, respectively, demonstrate anti-atherogenic and anti-calcific actions, and both of these effects are stronger in males than in females. In combination, the oral intake of these nutraceuticals might assist in controlling lipid and calcium profiles, at least in patients with an atherogenic lipid profile and high cardiovascular risk (e.g., because of multimorbidity). Notably, the male sex itself is a notable cardiovascular risk factor [116,117,118,119] that cannot be modified itself but can positively modulate various treatment effects, such as those of BCAA and MgCit. The start of the indicated supplementations at a young age may maximize their efficiency and extend the period of favorable lipid and mineral profiles. The verification of our results requires randomized controlled trials, including those comparing the performance of BCAA oral intake and intravenous injections of HSA with regard to serum albumin elevation, atheroprotection, and the lowering of serum calcification propensity. Different types of magnesium might also have a distinct efficiency in reducing serum ionized calcium and serum calcification propensity. Such studies may recruit multiple patient categories (not limited to cardiovascular disease and chronic kidney disease) for evaluating the potential beneficial effects of BCAA/HSA and various types of magnesium preparations. Considering other positive effects of BCAA and magnesium supplementations, the identification of their anti-atherogenic and anti-calcific effects may extend their application for the patients with high levels of atherogenic lipids (i.e., in a clinical setting reminiscent of the ApoE-knockout mouse model scenario), including those with familial hypercholesterolemia.

## 4. Materials and Methods

### 4.1. Animal Treatment Protocols

Male and female ApoE-knockout mice weighing 15–25 g, provided by the Research Institute for Complex Issues of Cardiovascular Diseases Core Facility, were used for all animal experiments (*n* = 270). Animals were allocated to polypropylene cages (one mouse per cage) lined with wood chips and had access to water and food (mouse chow with a 6% crude fat content) ad libitum. Throughout the duration of the experiment, the standard conditions of the temperature (24 ± 1 °C), relative humidity (55% ± 10%), and a 12 h light/dark cycle were carefully maintained, and the health status of all mice was monitored daily. No randomization was performed to allocate mice to experimental groups or cages. There were no specific inclusion or exclusion criteria. Experiments were conducted in a blinded fashion. All procedures were carried out conforming to the European Convention for the Protection of Vertebrate Animals used for Experimental and Other Scientific Purposes (Strasbourg, 1986) and were approved by the Local Ethical Committee of the Research Institute for Complex Issues of Cardiovascular Diseases (Kemerovo, Russia, ethical approval code: 5/1, date of approval: 10 April 2023).

Treatment regimens included the supplementation of drinking water with BCAA (55 mg/day, corresponding to a human equivalent dose of 13.5 g/day) or MgCit (1.85 mg/day, corresponding to a human equivalent dose of 450 mg/day). The human equivalent doses were calculated according to the respective guidelines [120] by dividing the dose for a mouse by 12.3 (a coefficient established by the Food and Drug Administration). Mock-treated mice received water without any supplementations. Such a dose of BCAA corresponded to the average of recommended range of BCAA intake (from 5 to 20 g/day) and to the recommended dose of magnesium intake (from 400 to 450 mg/day). A slight (≈10%) excess of BCAA and MgCit doses was made for the adjustment to the amounts of undrunk water. Water with or without BCAA and MgCit has been changed every 5 days.

To define the proper timing for the treatment initiation, the study included three independent experiments where treatment has been commenced at 1, 3, and 6 months of age. Each of three indicated experiments included 90 mice, equally distributed across the groups (30 animals receiving mock treatment, 30 animals receiving BCAA, and 30 animals receiving MgCit). To determine sex-specific effects of BCAA and MgCit interventions, each of the groups had equal samples of male and female mice (45 male and 45 female mice in total, 15 male and 15 female mice per group). In total, the study enrolled 270 mice (*n* = 90 per treatment group), including 135 male mice (*n* = 45 per treatment group) and 135 female mice (*n* = 45 per treatment group). Out of 270 mice, 222 (82.2%) survived to the study endpoint (i.e., 12 months of age). Survival rates were almost equal in male (113/135, 83.7%) and female (109/135, 80.8%, *p* = 0.52) mice. In the mock-, BCAA-, and MgCit-treated groups, respectively, 78/90 (86.7%), 76/90 (84.4), and 68/90 (75.5%) mice survived until the study endpoint (*p* = 0.12), and these rates did not differ significantly between male and female mice (mock-treated group: 41/45 (91.1%) and 37/45 (82.2%), respectively (*p* = 0.21); BCAA-treated group: 39/45 (86.7%) and 37/45 (82.2%), respectively (*p* = 0.56); MgCit-treated group: 33/45 (73.3%) and 35/45 (77.8%), respectively (*p* = 0.63)).

At the 12-month time point, heparin was injected (170 µL, 30 international units/mL) into the tail vein to prevent clotting; mice were euthanized by withdrawing the blood from the right renal vein under inhalation anesthesia (1% isoflurane, CAS 26675-46-7, Laboratorios Karizoo, Barcelona, Spain) and the aorta and left ventricle segment were excised with an AV. After the removal of perivascular adipose tissue and tunica adventitia, aortas were placed in a physiological saline solution (0.9% NaCl, CAS 7647-14-5, Solopharm, St. Petersburg, Russia) until further Oil Red O staining. AV-containing left ventricle segments were snap-frozen in a cryoembedding medium (K-FREEZE, AppScience Products, Moscow, Russia) at −80 °C until further sectioning.

### 4.2. Histological Analysis

Cryoembedded AVs underwent total cryosectioning (6 µm thickness, from 20 to 30 sections per mouse) and were stained with Oil Red O and Alizarin Red S to assess lipid retention and calcium deposition, respectively.

Oil Red O staining protocol of AV sections included their chemical fixation in 4% paraformaldehyde (CAS 30525-89-4, 029447, Central Drug House, New Delhi, India) for 10 min, three washes in phosphate-buffered saline (5 min per each, P071-1/B-60201, PanEco, Moscow, Russia), incubation in 60% isopropanol (CAS 67-63-0, HP-IS-AL01, ErgoProduction, St. Petersburg, Russia) for 5 min at room temperature, staining with 1.5% Oil Red O (CAS 1320-06-5, 23576, Sisco Research Laboratories, Mumbai, India) for 15 min at room temperature, three washes in 60% isopropanol (10 dips per each, CAS 67-63-0, HP-IS-AL01, ErgoProduction, St. Petersburg, Russia), counterstaining with Gill’s hematoxylin (CAS 517-28-2, HK-G0-BL01, ErgoProduction, St. Petersburg, Russia) for 10 min, bluing in a tap water for 5 min, and wash in double distilled water (10 dips). Coverslips were mounted using 50% aqueous glycerol solution (CAS 56-81-5, El010/042651, PanEco, Moscow, Russia). All procedures were conducted at room temperature.

Alizarin Red S staining protocol of AV sections included their chemical fixation in 95% ethanol (CAS 64-17-5, Kemerovo Pharmaceutical Plant, Kemerovo, Russia) for 30 min, three washes in double distilled water (10 s per each), staining with Alizarin Red S (CAS 130-22-3, 010043, LenReactiv, St. Petersburg, Russia) for 70 s, three washes double distilled water (10 s per each), and the mounting of coverslips using 50% aqueous glycerol solution (CAS 56-81-5, El010/042651, PanEco, Moscow, Russia). All procedures were conducted at room temperature.

Lipid retention and calcium deposition in the AVs was assessed using consecutive light and fluorescence microscopy (AxioImager.A1 microscope, EC Plan-Neofluar 20×/0.50 M27 objective, and Fs09—LP515 filter, Carl Zeiss, Oberkochen, Germany). Images were then evaluated in the ImageJ software (version 1.54k, National Institutes of Health, Bethesda, MD, USA) according to the following algorithm: demarcation of the AV leaflet on a light microscopy image by a freehand selection—transfer of the demarcated area to a fluorescence microscopy image (edit—selection—restore selection)—separation of the red channel (image—color—split channels)—manual threshold adjustment in the red channel if needed (image—adjust—threshold)—calculation of the proportion of Oil Red O- or Alizarin Red S-stained area (analyze—analyze particles—%area). The procedure was performed for all three AV leaflets, with the further calculation of an average proportion of Oil Red O- or Alizarin Red S-stained area.

Aortas (without removed tunica adventitia and perivascular adipose tissue) were opened longitudinally for en face staining, dehydrated in three changes of 85% isopropanol (5 min per each, CAS 67-63-0, HP-IS-AL01, ErgoProduction, St. Petersburg, Russia), stained in 1.5% Oil Red O (CAS 1320-06-5, 23576, Sisco Research Laboratories, Mumbai, India) for 16 h, and washed in three changes of 85% isopropanol (5 min per each, CAS 67-63-0, HP-IS-AL01, ErgoProduction, St. Petersburg, Russia) and three changes in phosphate-buffered saline (5 min per each, P071-1/B-60201, PanEco, Moscow, Russia). All procedures were conducted at room temperature. Lipid burden was assessed using stereomicroscopy (SZ51, Olympus, Tokyo, Japan). Images were then evaluated in the ImageJ software (version 1.54k, National Institutes of Health, Bethesda, MD, USA) according to the following algorithm: demarcation of the aorta by a freehand selection—analyze—measure—demarcation of each of lipid spots—analyze—measure—summation of lipid spot areas—calculation of the ratio between the total area of all lipid spots and area of the aorta. Stereomicroscopic images of the aorta were post-processed as in [121].

### 4.3. Measurement of Serum Ionized Calcium and Serum Calcification Propensity

The serum was obtained through centrifugation of the blood at 3000× *g* for 10 min (CM-6M, ELMI, Riga, Latvia). Serum ionized calcium was measured using an ion-selective electrode (Edan i15, Edan Instruments, Shenzhen, China). To assess whether HSA and MgCit affect serum calcification propensity, fresh serum was collected from the patients with STEMI (*n* = 100) admitted to the Research Institute for Complex Issues of Cardiovascular Diseases (Kemerovo, Russia). The study was conducted according to the Good Clinical Practice guidelines and the latest revision of the Declaration of Helsinki (2013) and was approved by the Local Ethical Committee of the Research Institute for Complex Issues of Cardiovascular Diseases (Kemerovo, Russia, ethical approval code: 8/1, date of approval: 3 July 2025). A written informed consent was provided by all study participants after receiving a full explanation of the study.

To evaluate serum calcification propensity, patient serum was supersaturated with CaCl_2_ (+2 mmol/L, CAS 10043-52-4, 8.06.01509, Khimreaktivsnab, Ufa, Russia) and Na_2_HPO_4_·12H_2_O (+2 mmol/L, CAS 10039-32-4, 8.06.00855, Khimreaktivsnab, Ufa, Russia) and subtracted OD_620_ values (INNO-S, LTek, Seongnam, Republic of Korea) from those of the paired control serum, which was diluted by double distilled water instead of CaCl_2_ and Na_2_HPO_4_·12H_2_O. The supersaturated samples contained 80 µL serum, 50 µL CaCl_2_ (+2 mmol/L, CAS 10043-52-4, 8.06.01509, Khimreaktivsnab, Ufa, Russia), 50 µL Na_2_HPO_4_·12H_2_O (+2 mmol/L, CAS 10039-32-4, 8.06.00855, Khimreaktivsnab, Ufa, Russia), and 20 µL physiological saline solution (0.9% NaCl, CAS 7647-14-5, Solopharm, St. Petersburg, Russia). The control samples contained 80 µL serum, 100 µL double distilled water, and 20 µL physiological saline solution. HSA (+10 mg/mL, 20 µL, Microgen, Moscow, Russia) or MgCit (+90 µg/mL, 20 µL, CAS 153531-96-5, 63067, Sigma-Aldrich, Saint Louis, MO, USA) was added instead of physiological saline solution in the treatment groups (both with and without Ca/P supersaturation). Therapeutic effect was calculated by subtracting OD_620_ values in the treatment groups from those of the supersaturated serum without HSA or MgCit. The specificity of such effect was verified by subtracting OD_620_ values in the treatment groups from those of the control serum without HSA or MgCit. In total, the experiment included eight groups (*n* = 100 patients and 800 OD_620_ measurements): (1) without CaCl_2_ and Na_2_HPO_4_·12H_2_O, HSA or MgCit; (2) without CaCl_2_ and Na_2_HPO_4_·12H_2_O but with HSA; (3) without CaCl_2_ and Na_2_HPO_4_·12H_2_O but with MgCit; (4) without CaCl_2_ and Na_2_HPO_4_·12H_2_O but with HSA and MgCit; (5) with CaCl_2_ and Na_2_HPO_4_·12H_2_O but without HSA and MgCit; (6) with CaCl_2_, Na_2_HPO_4_·12H_2_O, and HSA; (7) with CaCl_2_, Na_2_HPO_4_·12H_2_O, and MgCit; (8) with CaCl_2_, Na_2_HPO_4_·12H_2_O, HSA, and MgCit. The measurement of OD_620_ was conducted after 10 min of sample agitation on a microplate shaker (Titramax 1000, Heidolph, Schwabach, Germany) at 37 °C.

### 4.4. Statistical Analysis

Statistical analysis was performed using GraphPad Prism 8 (GraphPad Software, San Diego, CA, USA). The differences between the proportions were analyzed by a two-way chi-squared test with Yates’s correction for continuity. Data are presented as medians, 25th and 75th percentiles, and ranges. In the animal experiments, the values in three independent groups were compared using the Kruskal–Wallis test with Dunn’s multiple comparisons test. Paired measurements of serum calcification propensity in the patients with STEMI were analyzed using the Friedman test with Dunn’s multiple comparisons test. Adjusted *p* values ≤ 0.05 were regarded as statistically significant.

## 5. Conclusions

An early BCAA supplementation is capable of reducing lipid deposition in the AV, whilst MgCit is more efficient in decreasing serum ionized calcium. The indicated reduction in lipid burden in the AV and decline in serum ionized calcium were more pronounced in male than in female mice. The combined ex vivo treatment with HSA and MgCit curbed serum calcification propensity. BCAA intake can contribute to inhibiting the development of atherogenesis, at least in male patients, whereas MgCit supplementation holds benefits in regulating excessive serum calcium to reduce the corresponding cardiovascular risk. Our findings provide a rationale for the further randomized controlled trials of BCAA and MgCit to define their value for cardiovascular prevention.

## Figures and Tables

**Figure 1 ijms-26-11259-f001:**
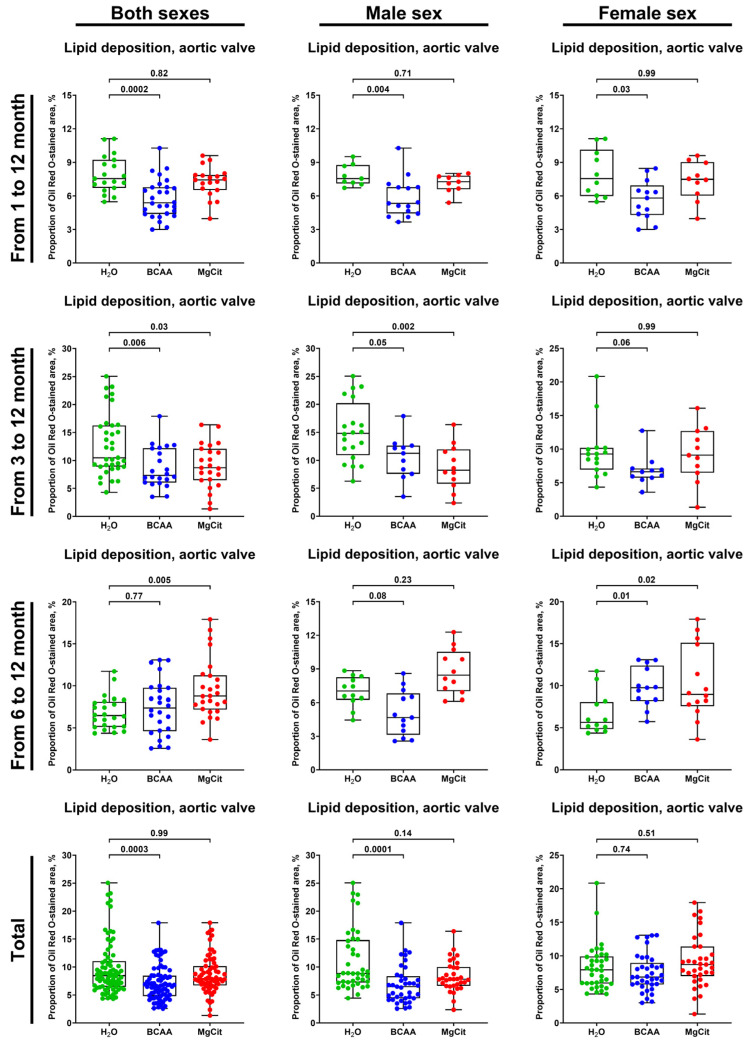
Semi-quantitative assessment of lipid deposition (measured through fluorescence microscopy of Oil Red O-stained sections) in the aortic valves of ApoE-knockout mice treated with proteinogenic branched-chain amino acids (BCAA, 55 mg/day, which corresponds to a human equivalent dose of 13.5 g/day) or magnesium citrate (MgCit, 1.85 mg/day, which corresponds to a human equivalent dose of 450 mg/day) from 1 to 12 months of age (11-month treatment duration), from 3 to 12 months of age (9-month treatment duration), or from 6 to 12 months of age (6-month treatment duration). Combined analysis of pooled samples is provided at the bottom. Green, blue, and red dots correspond to mock (H_2_O)-, BCAA-, and MgCit-treated mice, respectively. Each dot on the plots represents a measurement from one mouse. Whiskers indicate the range, box bounds indicate the 25th–75th percentiles, and center lines indicate the median. *p* values are provided above boxes; Kruskal–Wallis test with Dunn’s multiple comparisons test.

**Figure 2 ijms-26-11259-f002:**
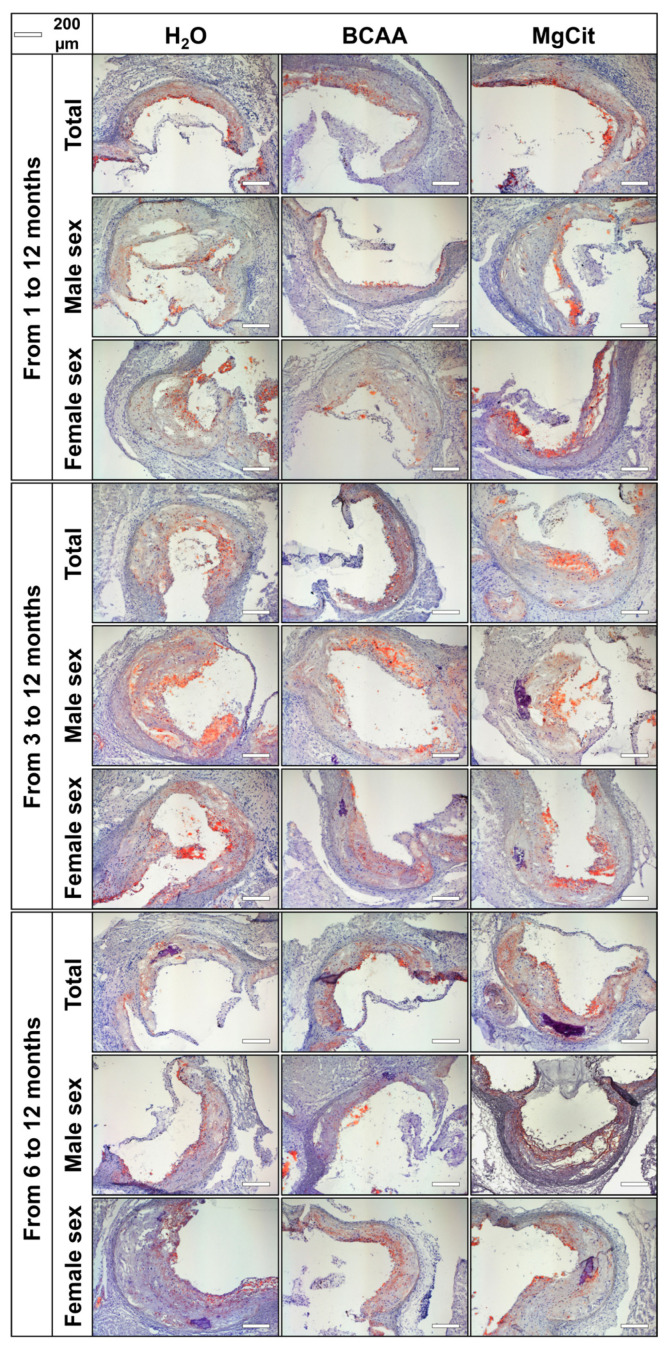
Aortic valves of ApoE-knockout mice treated with proteinogenic branched-chain amino acids (BCAA, 55 mg/day, which corresponds to a human equivalent dose of 13.5 g/day) or magnesium citrate (MgCit, 1.85 mg/day, which corresponds to a human equivalent dose of 450 mg/day) from 1 to 12 months of age (11-month treatment duration, top), from 3 to 12 months of age (9-month treatment duration, center), or from 6 to 12 months of age (6-month treatment duration, bottom). Mock-treated mice received water without any supplements. Representative light microscopy images of Oil Red O-stained aortic valve sections. Scale bar: 200 µm.

**Figure 3 ijms-26-11259-f003:**
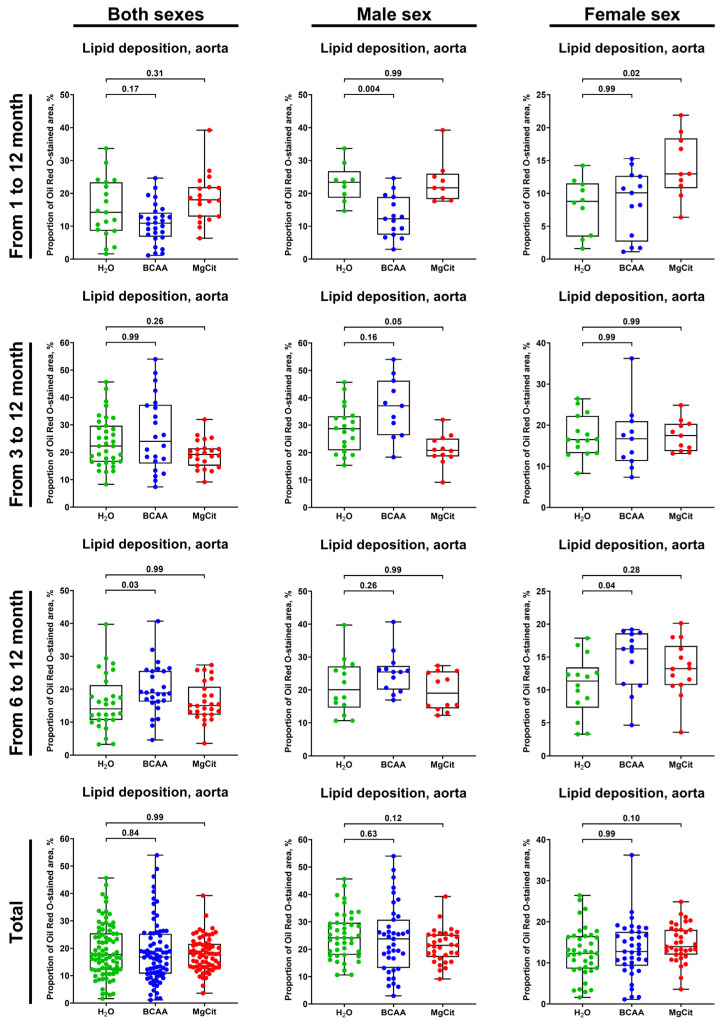
Semi-quantitative assessment of lipid deposition (measured by light microscopy following Oil Red O staining) in the aortas of ApoE-knockout mice treated with proteinogenic branched-chain amino acids (BCAA, 55 mg/day, which corresponds to a human equivalent dose of 13.5 g/day) or magnesium citrate (MgCit, 1.85 mg/day, which corresponds to a human equivalent dose of 450 mg/day) from 1 to 12 months of age (11-month treatment duration), from 3 to 12 months of age (9-month treatment duration), or from 6 to 12 months of age (6-month treatment duration). Combined analysis of pooled sample is provided at the bottom. Green, blue, and red dots correspond to mock (H_2_O)-, BCAA-, and MgCit-treated mice, respectively. Each dot on the plots represents a measurement from one mouse. Whiskers indicate the range, box bounds indicate the 25th–75th percentiles, and center lines indicate the median. *p* values are provided above boxes; Kruskal–Wallis test with Dunn’s multiple comparisons test.

**Figure 4 ijms-26-11259-f004:**
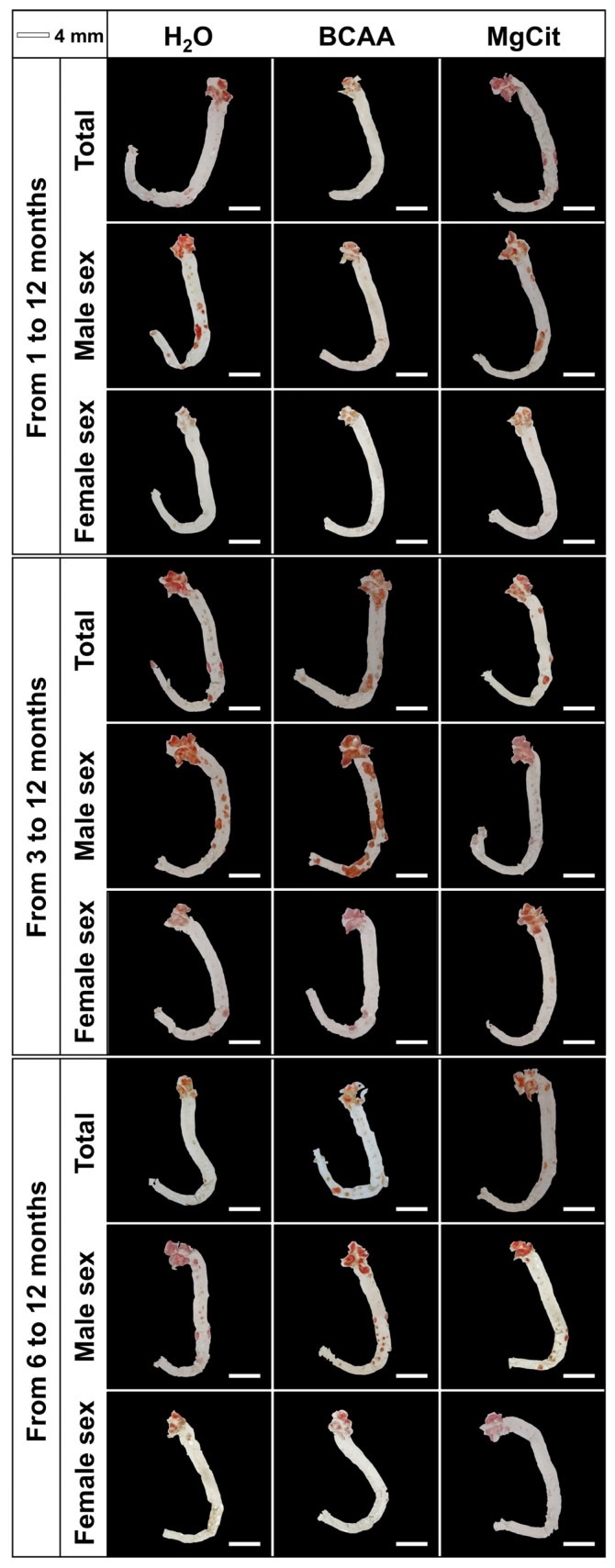
Aortas of ApoE-knockout mice treated with proteinogenic branched-chain amino acids (BCAA, 55 mg/day, which corresponds to a human equivalent dose of 13.5 g/day) or magnesium citrate (MgCit, 1.85 mg/day, which corresponds to a human equivalent dose of 450 mg/day) from 1 to 12 months of age (11-month treatment duration, top), from 3 to 12 months of age (9-month treatment duration, center), or from 6 to 12 months of age (6-month treatment duration, bottom). Mock-treated mice received water without any supplements. Representative en face images of Oil Red O-stained aortas. Scale bar: 4 mm.

**Figure 5 ijms-26-11259-f005:**
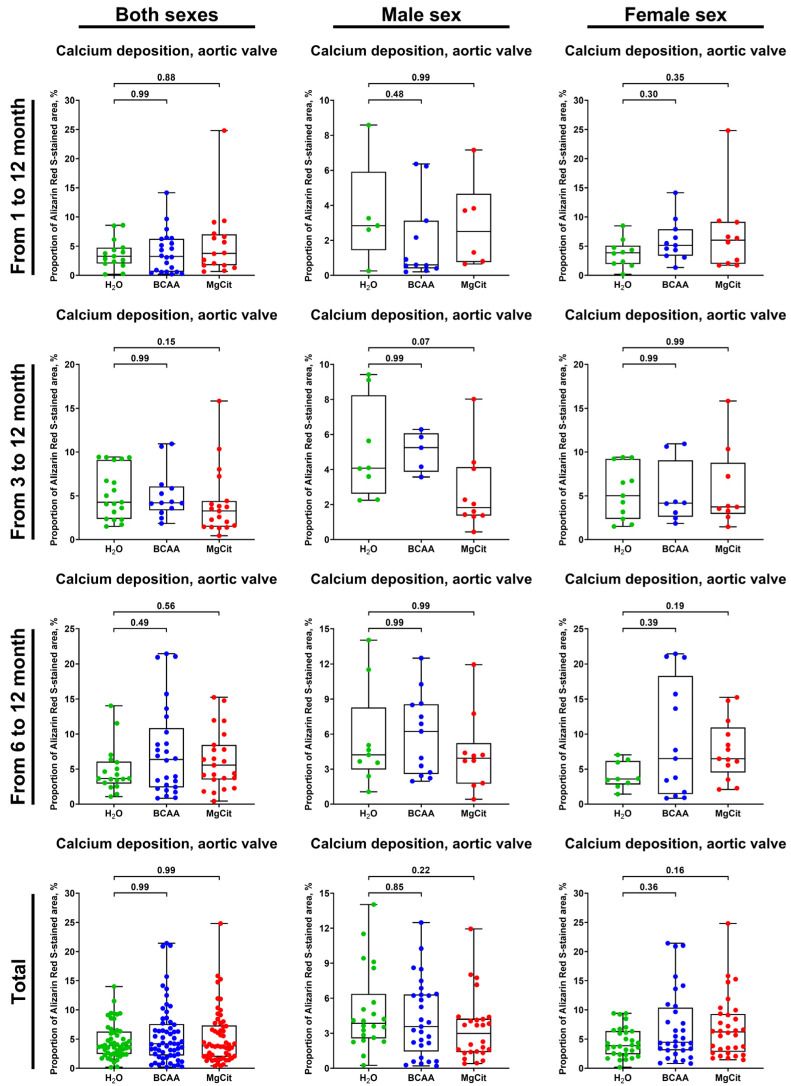
Semi-quantitative assessment of calcium deposition (measured by fluorescence microscopy of Alizarin Red S-stained sections) in the aortic valves of ApoE-knockout mice treated with proteinogenic branched-chain amino acids (BCAA, 55 mg/day, which corresponds to a human equivalent dose of 13.5 g/day) or magnesium citrate (MgCit, 1.85 mg/day, which corresponds to a human equivalent dose of 450 mg/day) from 1 to 12 months of age (11-month treatment duration), from 3 to 12 months of age (9-month treatment duration), or from 6 to 12 months of age (6-month treatment duration). Combined analysis of pooled samples is provided at the bottom. Green, blue, and red dots correspond to mock (H_2_O)-, BCAA-, and MgCit-treated mice, respectively. Each dot on the plots represents a measurement from one mouse. Whiskers indicate the range, box bounds indicate the 25th–75th percentiles, and center lines indicate the median. *p* values are provided above boxes; Kruskal–Wallis test with Dunn’s multiple comparisons test.

**Figure 6 ijms-26-11259-f006:**
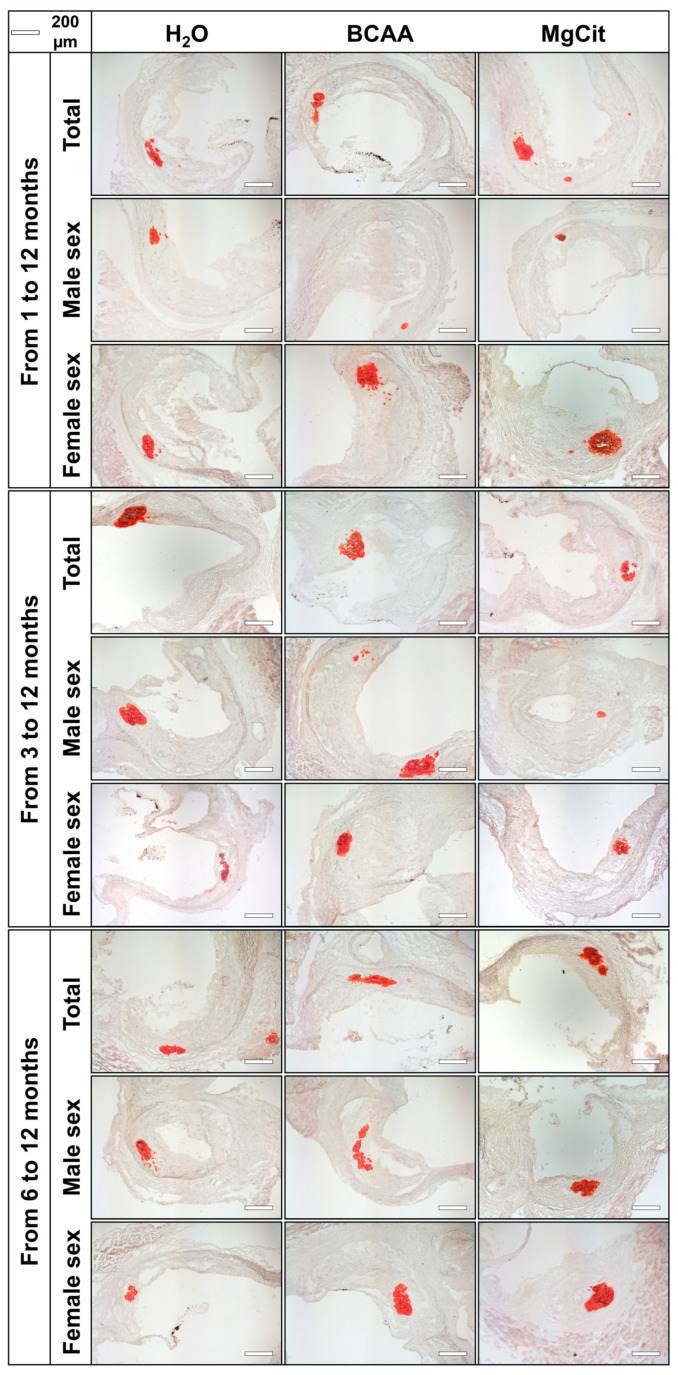
Aortic valves of ApoE-knockout mice treated with proteinogenic branched-chain amino acids (BCAA, 55 mg/day, which corresponds to a human equivalent dose of 13.5 g/day) or magnesium citrate (MgCit, 1.85 mg/day, which corresponds to a human equivalent dose of 450 mg/day) from 1 to 12 months of age (11-month treatment duration, top), from 3 to 12 months of age (9-month treatment duration, center), or from 6 to 12 months of age (6-month treatment duration, bottom). Mock-treated mice received water without any supplements. Representative light microscopy images of Alizarin Red S-stained aortic valve sections. Scale bar: 200 µm.

**Figure 7 ijms-26-11259-f007:**
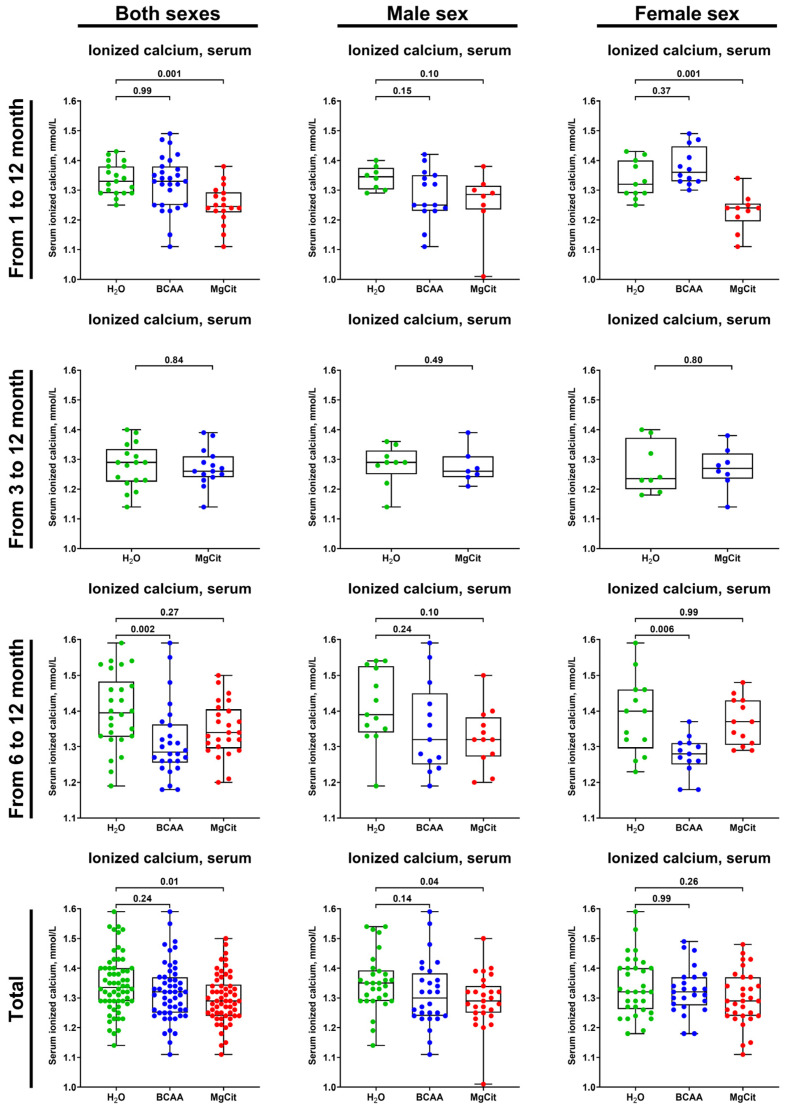
Measurement of serum ionized calcium (Ca^2+^) in of ApoE-knockout mice treated with proteinogenic branched-chain amino acids (BCAA, 55 mg/day, which corresponds to a human equivalent dose of 13.5 g/day) or magnesium citrate (MgCit, 1.85 mg/day, which corresponds to a human equivalent dose of 450 mg/day) from 1 to 12 months of age (11-month treatment duration), from 3 to 12 months of age (9-month treatment duration), or from 6 to 12 months of age (6-month treatment duration). Combined analysis of pooled sample is provided at the bottom. Green, blue, and red dots correspond to mock (H_2_O)-, BCAA-, and MgCit-treated mice, respectively. Each dot on the plots represents a measurement from one mouse. Whiskers indicate the range, box bounds indicate the 25th–75th percentiles, and center lines indicate the median. *p* values are provided above boxes; Kruskal–Wallis test with Dunn’s multiple comparisons test.

**Figure 8 ijms-26-11259-f008:**
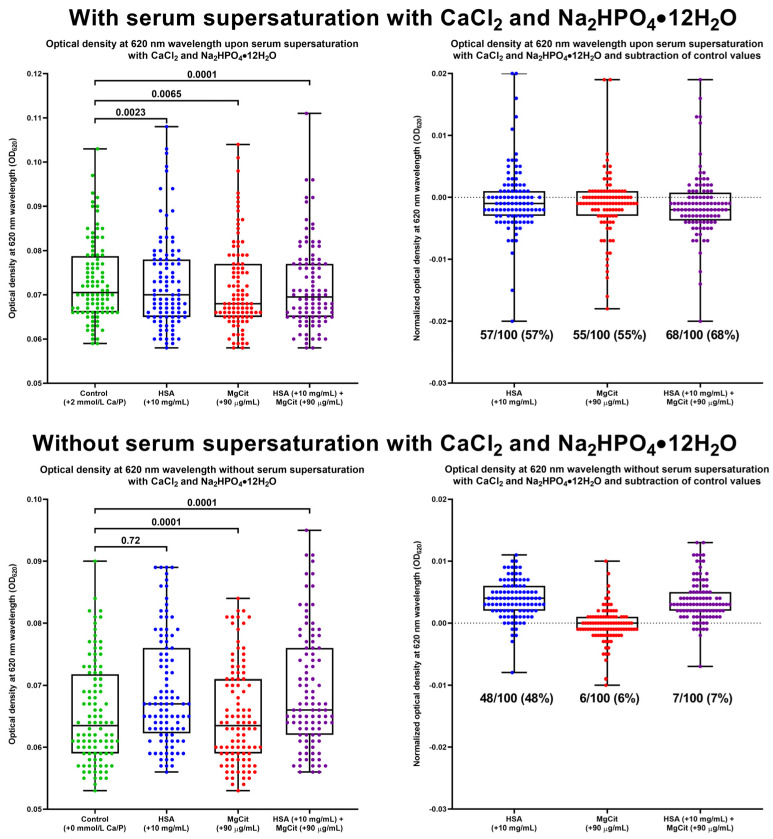
Assessment of serum calcification propensity following the addition of human serum albumin (HSA, +10 mg/mL), magnesium citrate (MgCit, +90 µg/mL), or both indicated solutions to the fresh serum of patients with ST-segment elevation myocardial infarction with (top) or without (bottom) supersaturated CaCl_2_ and Na_2_HPO_4_·12H_2_O (+2 mmol/L CaCl_2_ and +2 mmol/L Na_2_HPO_4_·12H_2_O). Serum calcification propensity was evaluated by measuring the optical density at 620 nm wavelength (OD_620_) in Ca/P-supersaturated and control serum after the 10 min incubation at 37 °C on a microplate shaker (**left**) and by subtracting OD_620_ values in the control serum from those in the Ca/P-supersaturated serum (i.e., normalized OD_620_, **right**). Green, blue, red, and violet dots correspond to control, HSA, MgCit, and combined HSA and MgCit treatments, respectively. Each dot on the plots represents a measurement from one patient. Whiskers indicate the range, box bounds indicate the 25th–75th percentiles, and center lines indicate the median. *p* values are provided above boxes; Friedman test with Dunn’s multiple comparisons test.

**Figure 9 ijms-26-11259-f009:**
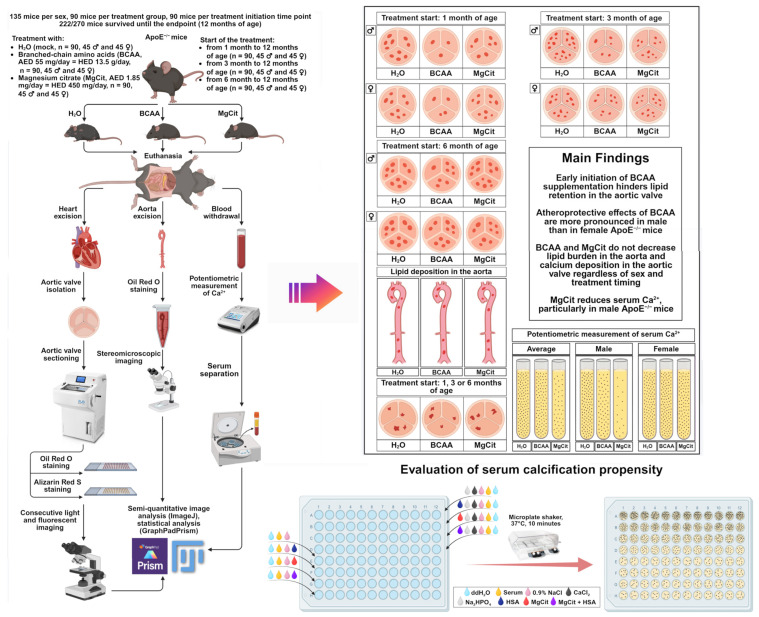
Study pipeline. The study included 270 ApoE-knockout mice: 135 mice per sex (male or female), 90 mice per treatment group (control H_2_O, proteinogenic branched-chain amino acids (BCAA), or magnesium citrate (MgCit)), and 90 mice per treatment initiation time point (1, 3, or 6 months of age). Of 270 mice, 222 survived until the endpoint (12 months of age). After the euthanasia, the aortic valves have been removed, sectioned, and stained with Oil Red O and Alizarin Red S to evaluate lipid and calcium deposition, respectively. Aortas were cleared of perivascular adipose tissue and tunica adventitia, stained with Oil Red O, and visualized by stereomicroscopic imaging. Semi-quantitative analysis of Oil Red O-stained and Alizarin Red S-stained images was performed using ImageJ software. Statistical analysis was conducted in GraphPad Prism. Serum has been withdrawn in order to conduct a potentiometric measurement of ionized calcium. Early BCAA intake (from 1/3 to 12 months of age) considerably reduced lipid retention in the aortic valve, whilst MgCit did not affect lipid deposition but decreased ionized calcium. Neither BCAA nor MgCit decreased lipid burden in the aorta or calcium deposition in the aortic valve. All indicated effects were more pronounced in male than in female mice. Furthermore, it was assessed whether human serum albumin (HSA) or MgCit could reduce serum calcification propensity in patients with ST-segment elevation myocardial infarction. Upon the mineral stress induced ex vivo (+2 mmol/L CaCl_2_ and +2 mmol/L Na_2_HPO_4_·12H_2_O) and 10 min incubation on the microplate shaker at 37 °C, HSA, MgCit, and particularly their combination decreased normalized OD_620_ (the difference in the OD_620_ between the supersaturated and the control serum samples).

## Data Availability

The raw data supporting the conclusions of this article will be made available by the authors on request.

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
