# Peer review of "Early Supplementation with Branched-Chain Amino Acids Ameliorates Lipid Retention in Aortic Valves of ApoE-Knockout Mice"

_ijms, 2025, doi:10.3390/ijms262311259_

Round 1
Reviewer 1 Report
Comments and Suggestions for Authors
The manuscript is promising and potentially suitable for publication. However, several drawbacks, including but not limited to the following, should be addressed to improve the quality of the manuscript before publication.
- You frequently report the mouse dose “corresponding to HED 13.5 g/day (BCAA) and 450 mg/day (MgCit).” Please add a one-sentence method on how HEDs were calculated.
- In Figure 4, the quality of the images could be further improved. The authors are encouraged to cite relevant literature (Figure 3B, J Cardiovasc Transl Res. 2025 Sep 10. doi: 10.1007/s12265-025-10637-8.) and revise the Oil Red O-stained aorta images accordingly.
- It would be beneficial if the authors could include a summary figure or schematic diagram to visually present and conclude the main findings.
- Remove the repeated sentence “All authors have read and agreed to the published version of the manuscript.”
Author Response
We sincerely thank the reviewer for the constructive criticism and valuable suggestions, which significantly helped us to improve the manuscript. We have provided a point-by-point response to the reviewer’s suggestions in the attached file. Please see the attachment.

Reviewer 2 Report
Comments and Suggestions for Authors
The manuscript entitled “Early supplementation with branched-chain amino acids ameliorates lipid retention in aortic valves of ApoE-knockout mice” evaluates the supplementation of male and female ApoE-knockout mice with BCAA (branched-chain amino acid) or magnesium citrate (MgCit) at different ages (1, 3, and 6 months up to 12 months of age) regarding lipid retention and calcium deposition in the aortic valve, lipid burden in the aorta, and serum ionized calcium. An ex vivo study with human serum albumin and MgCit in patients with myocardial infarction is also conducted regarding serum calcification propensity. The authors present the results and respective discussions, in addition to highlighting future studies that can be conducted to further address the gaps unaddressed in this study.
General observations
I recommend reviewing the passages in the text where it is written in the first person (using the word "we"), rewriting the sentences in the third person.
Review the use of acronyms such as HSA, BCAA, and MgCit in figures and captions without spelling them out in the caption the first time they are mentioned.
Abstract
Reading the abstract, at first, appears that the BCAA is human serum albumin. The study appears to focus on mice, but it states that tests were also conducted on patients.
The authors guide the reader well in the results and discussion sections, making it clear that the mouse study involved BCAA and MgCit, and that there was an additional patient study involving human serum albumin, which is synthesized from BCAA, and MgCit. However, this is not so clear in the abstract. I recommend that the authors revise this section to make it clearer to the reader what was done and what the results of each "step" were.
Keywords
As there are several proposed keywords, I suggest removing those already in the title, as this will not affect the article's subsequent searchability. The keywords "anti-atherogenic effect" and "anti-calcific effect" could be inserted.
Introduction
Lines 96-99 - The objective of the work can be supplemented with additional information to provide a more complete overview of the study. For example, there's no mention of the patient study. I recommend that the authors revise the objective to make clear to the reader what they found in this study.
Lines 99-106 - The information presented in these lines is the results of the work and should therefore be removed from the introduction. These results should be presented in the abstract, results, discussion and/or conclusion sections of the article. If the authors would like to better contextualize the topic, they can insert the hypothesis of the work at this point.
Author Response

(The authors gave the same response as above.)

Reviewer 3 Report
Comments and Suggestions for Authors
The paper presented to me for review is very well thought out and written. This is very interesting study on the effects of two nutraceuticals: branched-chain amino acids and magnesium citrate on reducing atherosclerotic plaque formation in anti-atherogenic and anti-calcific processes in ApoE-knockout mice. The paper presents a very detailed introduction with a description of the nutraceuticals used and the reason for their selection for the study. A strong point of the study is the large and equal number of male and female mice used in the experiments, which increased statistical power. The animals were divided into three time groups in terms of nutraceutical supplementation. Interestingly, the reduction in lipid retention at early supplementation with BCAA was higher in the aortic valves and aortas of male mice as compared to female mice. The authors add an important voice to the ongoing discussion on the gender differences in metabolism of BCAA and magnesium and response to BCAA and magnesium which could be affected by estrogen level (? Also my question).
I like the discussion in which the authors cite a lot of research. Overall, the paper cites a wealth of recent literature on the topic described.
Dear Authors, your work will undoubtedly be of interest of investigators, patients and physicians in the field of therapeutic strategies and the use of nutraceuticals in cardiovascular diseases prevention.
Author Response
We sincerely thank the reviewer for the high evaluation of our study. Indeed, the response to BCAA and magnesium citrate could be potentially affected by estrogens, and the impact of sex hormones on anti-atherogenic and anti-calcific effects of these supplementations (as well as on atherogenesis and calcification in ApoE-knockout mice) will be investigated in our further studies. Please see the attachment.

Round 2
Reviewer 1 Report
Comments and Suggestions for Authors
The authors have addressed the issues, and I suggest that the manuscript be accepted now.